# History-dependent switch-like differentiation of keratinocytes in response to skin barrier damage

**Elisa Domíguez-Hüttinger** [1]*, **Eliezer Flores-Garza**[2], **José Luis Caldú-Primo**[3], **Harley Day**[2], **Abihail Roque Ramírez**[4], **Reiko J. Tanaka** [2]*

**1** Departamento de Biología Molecular y Biotecnología, Instituto de Investigaciones Biomédicas, Universidad Nacional Autónoma de México, Ciudad Universitaria, Mexico City, México, **2** Department of Bioengineering, Imperial College London, South Kensington Campus, London, United Kingdom, **3** Doctorado en Ciencias Biomédicas, Universidad Nacional Autónoma de México, Ciudad Universitaria, Mexico City, Mexico, **4** Licenciatura en Física Biomédica, Facultad de Ciencias, Universidad Nacional Autónoma de México, Ciudad Universitaria, Mexico City, México

\* elisa.dominguez@iibiomedicas.unam.mx (EDH); r.tanaka@imperial.ac.uk (RJT)

## Abstract

The epidermis is formed by layers of keratinocytes with increasing levels of differentiation towards the outer skin called skin barrier, which protects our body from environmental stressors and dehydration. When skin barrier is damaged, keratinocyte differentiation is triggered, and terminally differentiated keratinocytes express skin barrier components, achieving skin barrier homeostasis. However, the dynamic and quantitative understanding of how skin barrier homeostasis is achieved remains unknown. To elucidate how keratinocyte differentiation is dynamically affected by skin barrier damage, especially in the presence of infection, we developed a mechanistic model of keratinocyte differentiation by integrating experimental results from 101 manually curated publications. To extract the key regulatory structure of the model, we applied model reduction, called the kernel reduction methodology to obtain the minimal reaction network. The key regulatory structure is characterised by positive feedback with cooperativity between Np63 and Stat3, two master regulators of keratinocyte differentiation. This regulatory structure gives rise to bistable behaviour for the expression of terminal differentiation markers of keratinocytes when the skin barrier is damaged and the extracellular calcium level is varied. We validated the model by confirming it produces the history-dependent and switch-like keratinocyte differentiation observed in in vitro reversibility assays. Analysis of the validated model shows that bacterial infection augments keratinocytes' sensitivity to skin barrier damage by decreasing the level required for differentiation and de-differentiation. Our results suggest the mechanisms by which skin barrier homeostasis is maintained even when the skin is exposed to fluctuating environments that perturb the barrier composition.

**Data availability statement:** The code and data are stored in our public GitHub repository: https://github.com/ElisaDominguezHuettinger/Keratinocyte_Differentiation.

**Funding:** EDH acknowledges funding from the Programa de Apoyo a Proyectos de Investigación e Innovación Tecnológica (PAPIIT) UNAM IA207822 (https://dgapa.unam.mx/index.php/impulso-a-la-investigacion/papiit) and from CONACyT Ciencia de Frontera 2022 (https://conahcyt.mx/ciencia-de-frontera/), project number 319600. The funders did not play any role in the study design, data collection and analysis, decision to publish, or preparation of the manuscript.

**Competing interests:** The authors have declared that no competing interests exist.

## Author summary

We propose and validate a mechanistic mathematical model that can uncover how keratinocyte differentiation is affected by skin barrier damage and infection. Our model represents the key regulatory structure of the complex network of biochemical interactions that map infectious microenvironments to keratinocyte differentiation states. We identify a *keratinocyte differentiation motif,* the key regulatory structure of the model, by applying systematic model reduction. The motif comprises positive feedback and cooperativity, which gives rise to a bistable dose-response behaviour for keratinocyte differentiation in response to skin barrier damage. We validate our model by confirming it reproduces the results of in vitro keratinocyte differentiation assays. Model analysis shows that innate immune responses triggered by infection decreases the threshold levels required for differentiation and de-differentiation, making keratinocyte differentiation more sensitive to skin barrier damage. These results help elucidate how infectious skin microenvironments trigger the dynamic regulation of keratinocyte differentiation and understand the role of infection in skin diseases such as eczema and psoriasis on epidermal barrier homeostasis.

## 1. Introduction

The epidermis is a stratified epithelial tissue formed by layers of keratinocytes with increasing levels of differentiation. Terminally differentiated keratinocytes in the uppermost layer of the skin constitute the skin barrier, a physical and chemical barrier that protects our body from environmental aggressors [1]. The skin barrier is maintained by feedback regulation of keratinocyte differentiation. This feedback regulation is triggered by skin barrier damage [2–7] via activation of transcription factors and signalling molecules that induce expression of terminal differentiation markers [8–15]. Keratinocyte differentiation is also modulated by infection, which often accompanies skin barrier impairment [16–21]. Skin barrier homeostasis is the ability of skin to maintain its barrier function despite external perturbations. Loss of skin barrier homeostasis is associated with impaired terminal differentiation of keratinocytes that results in the formation of a deficient skin barrier, leading to pathological phenotypes such as atopic dermatitis, psoriasis, and cutaneous squamous cell carcinoma [22–24] in which increased pathogen loads prevail.

Here, we aim to understand how infection and skin barrier damage contribute to skin barrier homeostasis by elucidating how keratinocyte differentiation is regulated by skin barrier damage, especially in the presence of infection.

Clarifying and predicting the relationship between keratinocyte differentiation, skin barrier damage, and infection is challenging from a purely empirical perspective due to the difficulties in performing quantitative experiments at a cellular level in a stratified multi-layered tissue, despite recent advances in in vitro epidermal or full-thickness skin models [25]. Systems biology approaches, in which experimental

data is integrated into predictive mathematical models, have proven to be effective in revealing the causal relationship between microenvironments and differentiation of, for example, T-cells [26], mesenchymal stem cells [26], and root stem cells [27]. However, mathematical models of dynamical processes for keratinocyte differentiation have been limited so far. Several regulatory networks of keratinocyte differentiation have been previously reconstructed at a cellular level using dynamical data from western blot experiments [28], public repositories [29], high throughput experiments [8] and, more recently, single-cell expression analysis [30,31]. However, these networks describe only the relationship between snapshot measurements without considering dynamical processes that are critical to elucidate causal relationships. As a result, the networks do not allow us to systematically analyse the effects of microenvironments on keratinocyte differentiation as they cannot reproduce the dynamical behaviour of keratinocyte differentiation.

In this paper, we propose a mechanistic model of keratinocyte differentiation that can predict the dynamic effect of microenvironments (characterised by levels and durations of skin barrier damage and infection) on keratinocyte differentiation. By integrating several experimental observations from published papers, the model describes how individual molecular players collectively contribute to keratinocyte differentiation, forming a regulatory network that maps microenvironments to keratinocytes' differentiation states. We further propose a *keratinocyte differentiation motif* by distilling the essential regulatory features of the regulatory network and show that the motif demonstrates a history-dependent switch-like differentiation of keratinocytes in response to skin barrier damage.

## 2. Results

### 2.1. Construction of a regulatory network for keratinocyte differentiation

We constructed a regulatory network for keratinocyte differentiation to investigate how microenvironments (levels and durations of skin barrier damage and infection) affect keratinocyte differentiation (Fig 1A). The network structure was determined by integrating the findings from 101 manually curated relevant publications (*Section A* in S1 Text and S1 Table).

The output of the network for keratinocyte differentiation is the expression level of Terminal Differentiation Markers (TDM) in keratinocytes, such as filaggrin (*Flg*), antimicrobial peptides (AMP), corneodesmosomes, and lipid processing enzymes. The high/low TDM expression level represents the differentiated/non-differentiated state of keratinocytes [32,33]. We consider the extracellular calcium level as the primary input of the regulatory network because its change is a major trigger of keratinocyte differentiation [15] as observed in in vitro calcium-switch experiments [34] and the extracellular calcium level rises in all epidermis layers upon skin barrier damage. The expression level of TDM [6,7] is altered by changes in the extracellular calcium level across the epidermis [35], especially for AMP [6,7], corneodesmosomes [28,36], and lipid processing enzymes [30,37]. As the second input of the regulatory network, we consider the concentration of active NFkB triggered by pathogen-induced innate immune responses, to evaluate the effects of infection on keratinocyte differentiation. Epidermal infection alters the calcium-mediated induction of keratinocyte differentiation [17–20,38–42] by interfering with the regulatory network.

The regulatory network for keratinocyte differentiation consists of 9 state variables: the Epidermal Growth Factor Receptor (EGFR), two AP1 transcription factors (cJun and JunB), p53, Np63, Notch, cMyc, miRNA203 and Stat3 (Fig 1B). These state variables are dynamically regulated with each other through transcriptional regulation, competitive inhibition, and post-translational and epigenetic modifications (detailed in *Section A* in S1 Text and in S1 Table).

To confirm that the regulatory network (Fig 1) robustly reproduces keratinocyte differentiation upon an increase in extracellular calcium levels, we formulate the network as an executable Boolean model (*Section B* and *Fig A* in S1 Text). This Boolean model allows us to describe the coupled dynamics of the nine state variables without requiring parameter values which are difficult to obtain for large systems. The model output is a discrete, quantitative and coarse-grained description of the all-or-nothing TDM response to an all-or-nothing calcium input. The deterministic steady states (fixed points and cyclic attractors) describe on/off patterns of the state variables that correspond to stable expression profiles triggered by different inputs.

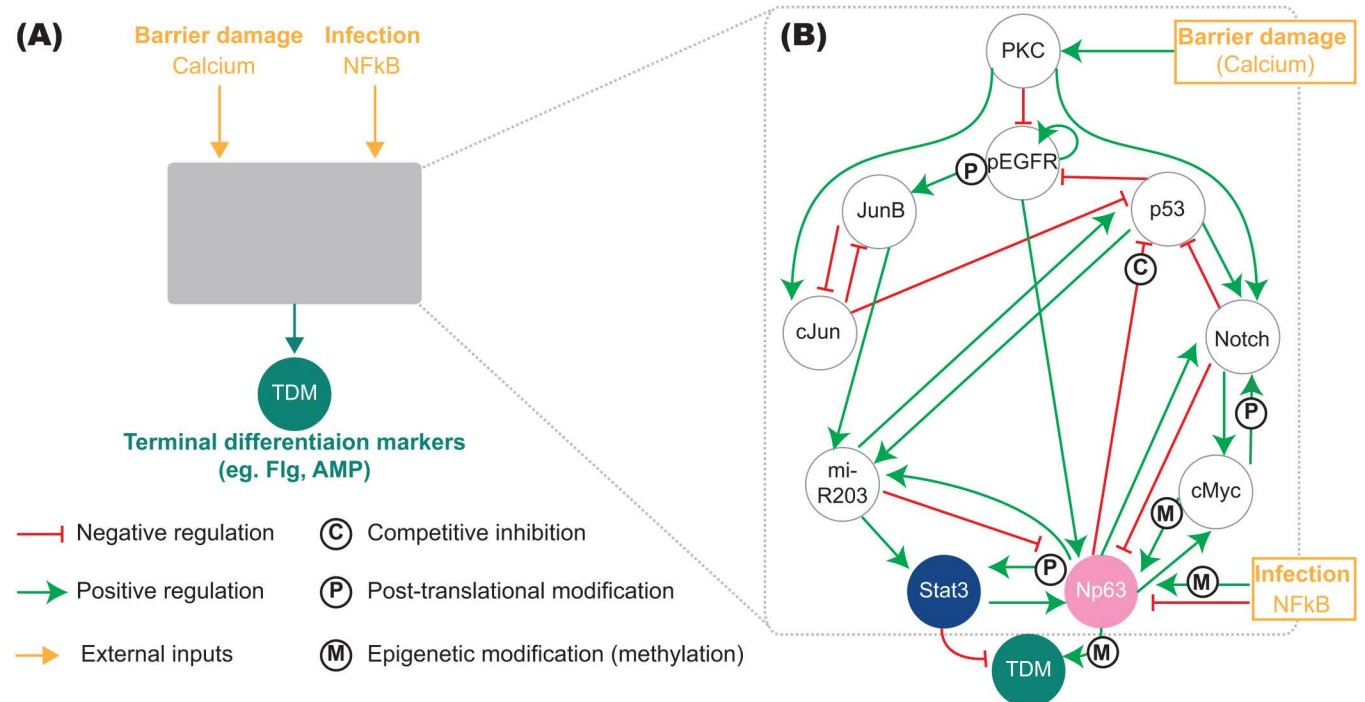

**Fig 1. Regulatory network for keratinocyte differentiation in response to the changes in the extracellular calcium level modulated by infection.** (A) Input-output relationship between microenvironmental signals and the expression of Terminal Differentiation Markers (TDM); (B) Regulatory network underlying keratinocyte differentiation. All regulations correspond to transcriptional events unless otherwise noted. PKC: Protein-Kinase C, EGFR: Epidermal Growth Factor Receptor.

The synchronous model simulation, under both low (basal) and high calcium conditions, finds four fixed point attractors, one of which corresponds to the differentiated state of keratinocytes with high TDM expression levels, and an additional two-state cyclic attractor that alternates between low and high TDM expression levels (*Fig A(i,ii)* in S1 Text). We confirmed that the four fixed point attractors are conserved under an asynchronous update regime (*Fig A(iv)* in S1 Text). The model has a much larger basin of attraction for the TDM state under the high, compared to the low (basal), calcium condition (*Fig A(iii)* in S1 Text), meaning that the differentiated state is much more likely to be observed under high calcium conditions (51.37% vs 7.42%). This result is consistent with that observed in calcium-switch experiments, where the population of differentiated keratinocytes increases dramatically upon increases in calcium [34].

## 2.2. Keratinocyte differentiation motif with positive feedback loops and cooperativity

The regulatory network (Fig 1) summarises most of the currently confirmed processes relevant to keratinocyte differentiation in response to skin barrier damage and infection. However, the network is too complex to fit to currently available experimental data (quantitative measurements of dynamic gene expression responses to inputs such as calcium and bacterial components) to quantitatively characterise how keratinocyte differentiation is affected by skin barrier damage and infection. We therefore reduced the network by applying the kernel reduction methodology [43] to obtain the minimal reaction network, which we refer to as the keratinocyte differentiation motif (Fig 2A). The kernel reduction is an algorithmic approach to identify the minimal essential network that preserves the input-output dynamics of the original network by sequentially removing intermediate nodes while keeping their regulatory interactions. We decided to remove all nodes of the regulatory network except for Stat3 and Np63. We kept Stat3 and Np63 because they directly regulate TDM, and their

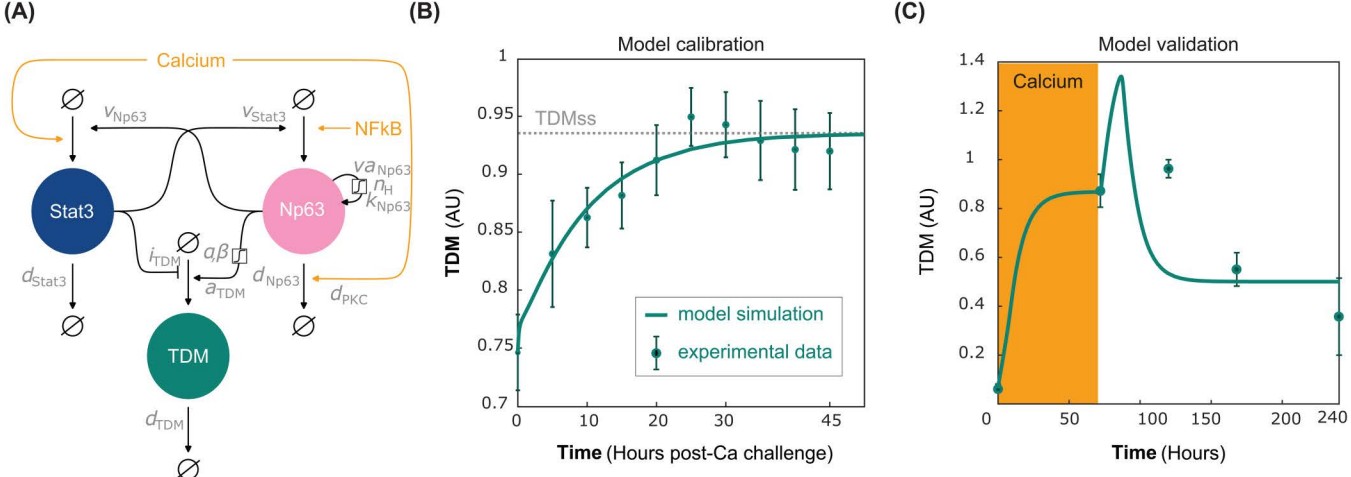

**Fig 2. The keratinocyte differentiation motif.** (A) Kinetic model. (B) Model fitting to TDM gene expression in response to a step increase in the calcium concentration (from 0.05 to 1.2 mM CaCl2); data from [8] represents the distribution (mean and standard deviation) over different terminal differentiation marker genes (SLPI, S100A7, and RNASE7, IVL and FLG), each of which was normalized by its maximal value. Error bars represent the dispersion (SD) of the individual genes. The steady state (TDMss) obtained by the model simulation is shown as a grey dotted line. (C) Validation of the kinetic model. The model reproduces the keratinocyte differentiation experiments from [44], in which a 3-day transient calcium challenge (shown in orange) is added to the medium and the reversion of keratinocyte differentiation is observed.

expression levels are measured in several calcium switch experiments. The details of the reduction process are described in *Section D* in S1 Text. We confirmed the minimality of the network as it robustly reproduces the experimentally observed dynamic and long-term responses to different pathological microenvironments, as detailed below, and this agreement with empirical observations is lost upon pruning further variables and interactions.

The keratinocyte differentiation motif consists of Stat3 and Np63 that directly regulate the TDM expression in response to the changes in the extracellular calcium level and infectious microenvironment (NFkB). An increase in the external calcium level (upon barrier damage) leads to an increase in Stat3 and a decrease in Np63 levels through molecular mechanisms described in *Section A* in S1 Text. Stat3 and Np63 increase each other's level, forming a positive feedback loop. Np63 expression is induced by cooperative auto-induction, forming the second positive feedback loop (Fig 2A). The TDM expression is induced by the increase in the Np63 expression level and inhibited by the activated Stat3. The expression of Np63 is induced by innate immune responses, represented here by the NFkB level.

The kinetics of the keratinocyte differentiation motif is described by

$$\frac{dStat3(t)}{dt} = Ca + v_{Np63} \cdot Np63(t) - d_{Stat3} \cdot Stat3(t),$$
(1)

$$\frac{dNp63(t)}{dt} = va_{Np63} \cdot \frac{Np63(t)^{n_H}}{k_{Np63}{}^{n_H} + Np63(t)^{n_H}} + v_{Stat3} \cdot Stat3(t) + NFkB - Np63(t) \cdot (d_{Np63} + d_{Ca} \cdot Ca)$$
(2)

$$\frac{dTDM(t)}{dt} = \frac{a_{TDM} \cdot Np63(t) * e^{-\beta t}}{1 + i_{TDM} \, Stat3(t)} - d_{TDM} \cdot TDM(t),$$
(3)

where $Stat3(t)$, $Np63(t)$ and $TDM(t)$ represent the expression levels of Stat3, Np63 and TDM, respectively. The direct regulatory interactions are modelled by the law of mass action kinetics. Stat3 expression is induced by calcium with

a constant rate, $Ca$, which is a lumped parameter representing the effects of calcium on the dynamics of Stat3, and by Np63 with a rate, $v_{Np63}$. We describe the auto-induction of Np63 expression by a Hill function (with a maximal rate, $va_{Np63}$, a half-maximal inductor concentration, $k_{Np63}$, and a Hill-coefficient, $n_H$) because it is mediated by the formation of protein complexes. Induction of Np63 expression via a Stat3-dependent and NFkB-dependent pathways are described with rates, $v_{Stat3}$ and $NFkB$, respectively. Expression of TDM is inhibited by Stat3 with rate, $i_{TDM}$, and is augmented by Np63 methylation of their promoters [45–47]. It is modelled with the convolution of $Np63(t)$ with a decaying exponential ($a_{TDM} \cdot Np63(t) * e^{-\beta t}$) with maximal rate, $a_{EDC}$, and the exponent, $\beta$, quantifying the memory of the methylation. The natural degradation of Stat3, Np63 and TDM are described by the rates, $d_{Stat3}$, $d_{Np63}$, and $d_{TDM}$, respectively. Np63 degradation is also calcium-dependent with a weighting, $d_{Ca}$. We obtained the kinetic model parameters by minimising the difference between the model simulation and the dynamic experimental data from primary human keratinocytes [8] using a global optimisation algorithm (Table 1). The fitted model captures a slow steady increase of TDM expression in keratinocytes (AMP SLPI, S100A7, RNASE, filaggrin and involucrin) [8] for 48h upon calcium challenge (Fig 2B and Fig C in S1 Text).

We validated our model with calibrated parameters by confirming that the model dynamics reflect the qualitative dynamical behaviour of the reversible keratinocyte differentiation assays [44] (Fig 2C), the decaying dynamics of Np63 expression observed in two independent experiments [8,48] (*Fig D* in S1 Text), and the dynamics of filaggrin expression observed in a keratinocyte differentiation assay in human normal epidermal keratinocytes [36] (*Fig E* in S1 Text).

**Table 1. Nominal parameters of the kinetic model.**

| Symbol | Nominal values | Units | Description |
|---|---|---|---|
| Inputs | | | |
| **Ca** | 0.1 (low) to 5 (high) | [a.u.]/[t] | Ca-mediated induction of Stat3 |
| **NFkB** | 0 | [a.u.]/[t] | Infection level-mediated induction of Np63 |
| Stat3-specific parameters | | | |
| $v_{Np63}$ | 2 | $1/[t]$ | Rate of Stat3 production induced by Np63 (positive feedback #1) |
| $d_{Stat3}$ | 1 | $1/[t]$ | Stat3 degradation rate |
| Np63-specific parameters | | | |
| $va_{Np63}$ | 10 | $1/[t]$ | Maximal rate of Np63 production induced by Np63 (positive feedback #2) |
| $v_{Stat3}$ | 1 | $1/[t]$ | Rate of infection-mediated production of Np63 |
| $n_H$ | 3 | NA | Hill-coefficient for the positive feedback of Np63 on Np63 |
| $k_{Np63}$ | 1.35 | $[y]$ | Half-maximal Np63 concentration for Np63 production |
| $v_{Stat3}$ | 1 | $1/[t]$ | Rate of Np63 production induced by Stat3 (positive feedback #3) |
| $d_{Np63}$ | 6 | $1/[t]$ | Natural degradation rate of Np63 |
| $d_{Ca}$ | 0.5 | $1/[y]$ | Rate of Ca-mediated Np63 degradation |
| Epidermal differentiation markers-specific parameters | | | |
| $a_{TDM}$ | $68 \times 10^3$ | $1/[t]$ | Rate of TDM expression induced by Np63 |
| $i_{TDM}$ | 500 | $1/[y]$ | Rate of Stat3-mediated inhibition of TDM expression |
| $d_{TDM}$ | 0.1 | $1/[t]$ | TDM degradation rate |
| $\beta$ | 451 | $1/[t]$ | Decaying exponential for the effects of Np63 on TDM expression |

## 2.3. Switch-like and history-dependent keratinocyte differentiation in response to change in the extracellular calcium level

We demonstrated that the kinetic model of keratinocyte differentiation shows a bistable behaviour by deriving the null-clines of the 2D Np63-Stat3 projection of the model (*Section E* in S1 Text). The two stable steady states in our model are visualised as the points of intersection between the Stat3 nullcline (a first-order polynomial) and the Np63 nullcline (a sigmoidal function) (Fig 3A). The effects of changing the calcium level (the primary input of our model) on the stable steady states are visualised on the Np63-Stat3 phase plane, onto which the state trajectories of the model can be projected as these two state variables are uncoupled from the output (TDM). We investigated the steady state behaviour of the model under low and high calcium concentrations by simulating varying levels of CaCl2 [mM], where the low calcium concentration corresponds to that typically used in in vitro calcium switch experiments, and the high concentration is one order of magnitude higher (Fig 3A). As the calcium level increases, the Stat3 nullcline shifts upwards while the Np63 nullcline straightens, eventually losing the low Np63 steady state. As a result, only the undifferentiated state with low Np63 is stable for low calcium conditions, while two steady states, corresponding to a low and a high Np63 (and Stat3), exist for medium calcium conditions.

Bifurcation analysis confirms the bistability for Np63, Stat3 and TDM with the calcium level as a bifurcation parameter between the threshold calcium concentrations, $C-$ and $C+$ (Fig 3B). Increasing calcium concentration above the threshold, $C+$, results in an abrupt increase in the levels of the state variables, which persists until the calcium concentration is decreased to a value below the second threshold, $C-$, at which an abrupt change from high to low is observed. The bistability is consistent with a reversible switch-like history-dependent keratinocyte differentiation observed in experimental studies in response to a change in the extracellular calcium level. For example, the TDM expression levels are stabilised eventually at a low or high state [36,49,50] after a transient change triggered by a change in the extracellular calcium level. When stabilised at a high state due to a sufficiently long-lasting increase in the calcium level, it remains at a high state for days after the extracellular calcium level decreases [44].

## 2.4. Modulation of keratinocyte differentiation by infection

To investigate how the infectious microenvironment modulates keratinocyte differentiation, we conducted a similar bifurcation analysis but with an increased level of active NFkB to mimic pathogen-induced innate immune responses (Fig 4). In

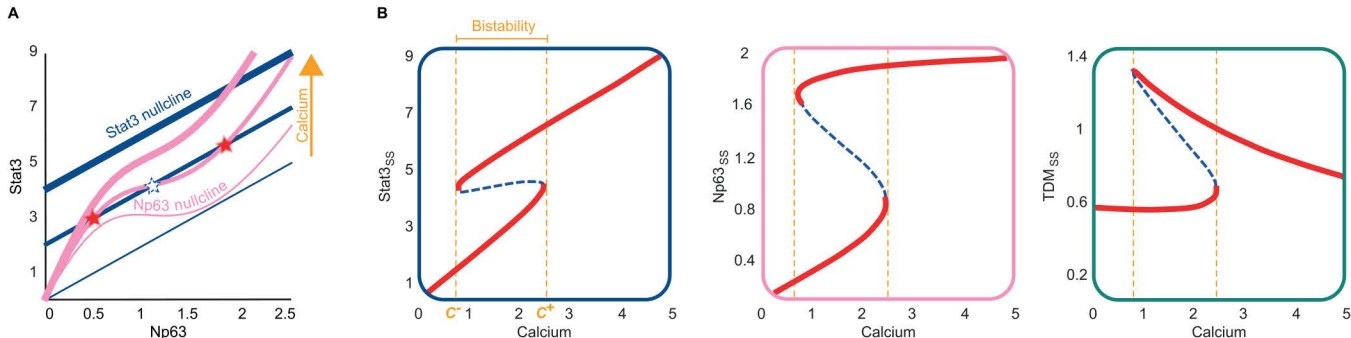

**Fig 3. Bistability observed in the kinetic model.** (A) The Np63-Stat3 phase plane. Three intersection points of the Np63 and Stat3 nullclines for intermediate calcium levels correspond to two stable (filled stars) and one unstable (open blue star) steady states. Increasing the calcium levels shifts the Stat3 nullcline up and straightens the Np63 nullcline, leading to the loss of the low stable steady state. (B) Bifurcation diagrams for the three state variables as the extracellular calcium level as a bifurcation parameter. Bistabliltiy is observed between the threshold calcium concentrations $C-$ and $C+$, at which an abrupt change in the state variables is observed.

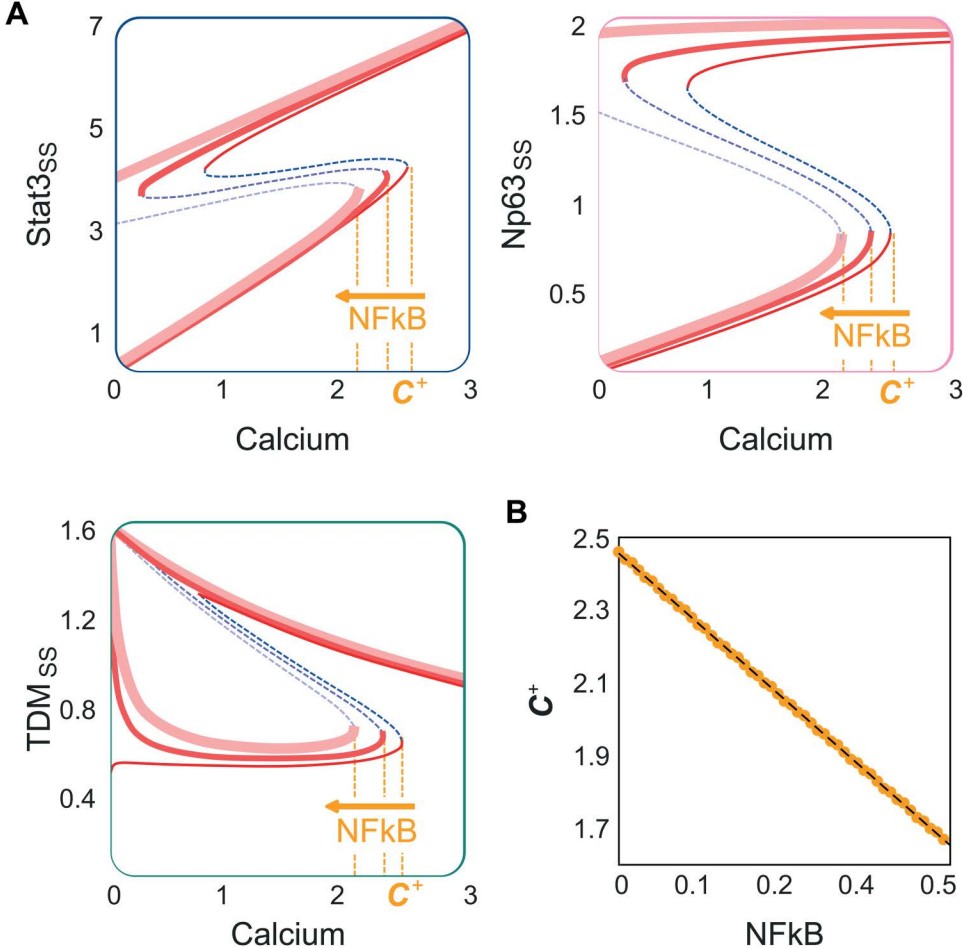

**Fig 4. Effect of infection-induced immune response (NFkB) on keratinocyte differentiation motif.** (A) Bifurcation diagrams for the 3 state variables (Stat3, Np63 and TDM) with three levels of NFkB (0, 0.1 and 0.25). The larger the NFkB level is, the smaller the activation threshold level for the calcium (C+), making the system more sensitive to barrier damage. (B) C+ linearly decreases with NFkB (variation of NFkB: 0:0.01:1).

the keratinocyte differentiation motif, an increase in the level of active NFkB leads to an increase in the Np63 production rate as NFkB increases the transcription rate of Np63 by inducing activation of p300 [51–53].

Increasing NFkB in our model lowers the activation thresholds of calcium *(C+)* required for keratinocyte differentiation. A lower threshold represents the skin being more sensitive to barrier damage: more subtle barrier damage (a slight increase in the extracellular calcium level) is sufficient to trigger the TDM expression in the presence of NFkB. Small and transient barrier damage can lead to a burst in TDM. This result is consistent with the increased sensitivity of keratinocyte differentiation to barrier damage in the presence of pathogens [17–20,38–40,54].

## 3. Discussion

This paper proposed the first mechanistic model of keratinocyte differentiation. The model development comprised of network assembly, simulation, and validation of both the full and the minimal regulatory networks of keratinocyte differentiation.

To develop the mechanistic model, we first conducted an extensive literature search and assembled a regulatory network of intercellular interactions involved in keratinocyte differentiation. We then confirmed that calcium-triggered

keratinocyte differentiation emerges from this network by dynamically simulating it using a Boolean network approach. As the whole network is too complex to analyse its dynamics, we derived the minimal regulatory network of keratinocyte differentiation by identifying the smallest set of variables and their interactions that can robustly reproduce the experimentally observed keratinocyte differentiation in response to calcium and infection. The resulting keratinocyte differentiation motif is then mathematically represented using kinetic ordinary differential equations. Model parameter values were obtained by global optimisation to fit the model to time-course data of calcium switch experiments. Bifurcation analysis of our model reproduces the abrupt history-dependent keratinocyte differentiation in response to the changes in the calcium level reported experimentally. Bifurcation analysis also showed that infection-induced immune responses shape the decision-making process of keratinocyte differentiation by shifting the extracellular calcium level thresholds required for stable TDM expression.

Our proposed keratinocyte differentiation motif comprises the smallest set of variables and their interactions that can reproduce various empirical observations of keratinocyte differentiation in response to calcium and infection derived from different experimental conditions. Adding more nodes could improve the model's ability to capture more experimental results, including the deleterious effects of HPV [42,49,55] and inflammation [54,56,57] on keratinocyte differentiation. However, it would also add more parameters to the model. Given the scarcity of available quantitative and longitudinal data, obtaining reliable estimates for those parameters would be difficult. We hope our work will motivate experimentalists to generate more quantitative time-resolved measurements of keratinocyte differentiation.

In summary, our mathematical model analysis uncovered the keratinocyte differentiation motif, comprised of the interplay between Stat3 and Np63, as the key regulatory structure underlying keratinocyte differentiation in response to the changes in the extracellular calcium level. The response is modulated through infection-induced immune responses, as infection increases the sensitivity to calcium-mediated increase in TDM expression by increasing Np63 production.

Our work contributes to elucidating the decision-making processes underlying keratinocyte differentiation [15] and its role in shaping the homeostasis of the epidermis and other stratified epithelial tissues. Skin barrier homeostasis has been previously modelled using multi-scale models [58–60], which however do not consider the mechanisms of keratinocyte differentiation. The proposed minimal network of keratinocyte differentiation is mechanistic yet simple enough to be incorporated into such a multi-scale model of epidermal dynamics. It will be interesting to analyse the contribution of the tissue-level feedback from skin barrier function to keratinocyte differentiation and epidermal homeostasis and elucidate how the differentiation state of keratinocytes affects the immune response to pathogens (secretion of AMP) and barrier restoration in response to barrier damage and pathogen challenge. Such a model would contribute to the understanding of the mechanisms through which treatments to enhance keratinocyte differentiation directly (e.g., vitamin D [61]) or indirectly through interference with IL4 signalling (Dupilumab [62,63]) help the restoration of epidermal homeostasis in diseases such as atopic dermatitis and psoriasis.

## 4. Methods

### 4.1. Curation of dynamic data for epidermal differentiation markers

We assembled expression data of mRNAs (measured by qPCR and by microarray) and proteins (measured by Western Blot) from 14 references (S2 Table) to test the validity of our kinetic model of keratinocyte differentiation. It includes time-course data of the TDM (involucrin, fillagrin, transglutaminase [36,49,50,64], AMP HBD [7] and the internal regulators of epidermal differentiation (ΔNp63 [48] and pEGFR, cMyc and cJun [28]) in response to calcium challenges (a sudden increase from 0.05mM to 1.2mM or 1.3mM CaCl) under control conditions and inflammatory [56,64,65] or TLR-activating [16] microenvironmental conditions, as well as a reversibility experiment [44] through which the memory of keratinocyte differentiation can be quantitatively assessed.

### 4.2. Parameter optimisation of the kinetic model for keratinocyte differentiation

We used the GlobalSearch function in Matlab R2022a to minimise the difference between predicted and experimentally determined mean-over individual gene expression of the TDM: SLPI, S100A7 RNASE (AMP), and filaggrin and involucrin measured by Toufighi et al. [8]. Calcium switch experiments were simulated by increasing the values of Ca from 0.1 to 2.

### 4.3. Bifurcation analysis

Steady states were computed numerically using vpasolve function in Matlab R2022a, and their stability was evaluated by assessing the sign of the eigenvectors of the corresponding Jacobian matrix.

## Supporting information

**S1 Table. Individual regulatory interactions underlying the regulatory network for keratinocyte differentiation in response to the changes in the extracellular calcium level modulated by infection assembled from 101 references.**
(XLSX)

**S2 Table. Expression data of epidermal differentiation markers corresponding to levels of mRNAs (measured by qPCR and by microarray) or proteins (measured by Western Blot) assembled from 14 references.**
(XLSX)

**S1 Text. The Supplementary Text contains Supplementary *Sections A-F*, Supplementary *Figures A-F* and Supplementary References.**
(PDF)

## Author contributions

**Conceptualization:** Elisa Domínguez-Hüttinger, Reiko J Tanaka.

**Data curation:** Elisa Domínguez-Hüttinger, Eliezer Flores-Garza, José Luis Caldú-Primo, Abihail Roque Ramírez, Reiko J Tanaka.

**Formal analysis:** Elisa Domínguez-Hüttinger, Eliezer Flores-Garza.

**Funding acquisition:** Elisa Domínguez-Hüttinger, Reiko J Tanaka.

**Investigation:** Harley Day.

**Methodology:** Elisa Domínguez-Hüttinger, Reiko J Tanaka.

**Project administration:** Elisa Domínguez-Hüttinger, Reiko J Tanaka.

**Software:** Elisa Domínguez-Hüttinger, Eliezer Flores-Garza, Abihail Roque Ramírez.

**Supervision:** Elisa Domínguez-Hüttinger, Reiko J Tanaka.

**Visualization:** Elisa Domínguez-Hüttinger, Eliezer Flores-Garza, Reiko J Tanaka.

**Writing – original draft:** Elisa Domínguez-Hüttinger, Reiko J Tanaka.

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
