## [Decision Letter · Decision Letter 0]

PCOMPBIOL-D-24-01530

History-dependent switch-like differentiation of keratinocytes in response to skin barrier damage

PLOS Computational Biology

Dear Dr. Domínguez Hüttinger,

Thank you for submitting your manuscript to PLOS Computational Biology. After careful consideration, we feel that it has merit but does not fully meet PLOS Computational Biology's publication criteria as it currently stands. Therefore, we invite you to submit a revised version of the manuscript that addresses the points raised during the review process.

Please submit your revised manuscript within 60 days Mar 16 2025 11:59PM. If you will need more time than this to complete your revisions, please reply to this message or contact the journal office at ploscompbiol@plos.org. Please include the following items when submitting your revised manuscript:

We look forward to receiving your revised manuscript.

Kind regards,

Ricardo Martinez-Garcia

Academic Editor

PLOS Computational Biology

Stacey Finley

Section Editor

PLOS Computational Biology

**Additional Editor Comments :**

The reviewers raise several points related to details about the model. These issues should be justified and/or clarified to strengthen the paper.

**Journal Requirements:**

At this stage, the following Authors/Authors require contributions: José Luis Caldú-Primo. Please ensure that the full contributions of each author are acknowledged in the "Add/Edit/Remove Authors" section of our submission form.

3) We noticed that you used the phrase 'not shown' in the manuscript. We do not allow these references, as the PLOS data access policy requires that all data be either published with the manuscript or made available in a publicly accessible database. Please amend the supplementary material to include the referenced data or remove the references.

5) We have noticed that you have uploaded Supporting Information files, but you have not included a list of legends. Please add a full list of legends for your Supporting Information files after the references list.

6) Please ensure that the funders and grant numbers match between the Financial Disclosure field and the Funding Information tab in your submission form. Note that the funders must be provided in the same order in both places as well. Currently, the order of the funders is different in both places.

Please indicate by return email the full and correct funding information for your study and confirm the order in which funding contributions should appear. Please be sure to indicate whether the funders played any role in the study design, data collection and analysis, decision to publish, or preparation of the manuscript.

**Reviewers' comments:**

Reviewer's Responses to Questions

Reviewer #1: In this manuscript, the authors have put together a regulatory network for keratinocyte differentiation and performed Boolean modelling on this network to check if it matches experimental patterns. Then, they performed model reduction to get a minimal motif and performed bifurcation analyses to simulate conditions such as skin barrier damage and infection through Ca and NFkB, respectively. The efforts of the authors in putting together a network that captures keratinocyte differentiation mechanism should be appreciated. However, this study has many major concerns still:

1. It is unclear how many curated publications were used to construct the network: mentioned to be 96 in Line 28 of the abstract, but 85 in Line 120 of the main text, and 85 in Supplementary section S1.

2. How the inputs and outputs are connected in Figure 1B is not clear, and it would be great if these edges could be added. Additionally, it would be helpful if this network could be provided in a Supplementary table, maybe as an edge list format: source-node, target-node, effect, references.

3. What is the reason for the authors to include the boolean simulations in the supplementary section?

4. The presence of cyclic attractors is a bit concerning regarding the interpretation, more so in the High Ca condition. Does this mean that the cell switches back and forth between differentiated and undifferentiated states? Generally, asynchronous boolean tends to have a loss of such cyclic attractors; the authors can perform these simulations and verify if the results of fixed points remain consistent.

5. I also find the model reduction to be unsatisfactory. For the interaction from Np63 to Stat3, apart from the direct edge and indirect edges via miR-203, the effective regulation through other indirect edges seems to be mostly -ve. Can the authors find the effective regulation between Stat3 and Np63 over the indirect paths and justify the regulations they have considered in the current model? In fact, in Section S4, the model reduction gives a higher number of negative interactions from Np63 to Stat3. The authors comment, "For the incoherent regulation (2), we decided to keep only the positive effect of Np63 on Stat3 (Np63 induces Stat3) because adding the negative effect in a mathematical model did not alter the main features of our interest, including observation of the bistable behaviour and the fit to data." This justification seems very unsatisfactory.

6. The validation provided in Figure 2C is also not satisfactory. The peak just after Ca challenge withdrawal seems like an artefact of the model and not a biological trend. However, this cannot be said for sure, given the lack of temporal resolution of data. Regardless, the fit to the rest of the data points is visually bad, and this can be objectively measured by any goodness of fit metric.

7. The authors should explain their choice of hills functions specifically for the autoinduction of Np63 while using linear terms for the rest of the regulations. Additionally, there seems to be a missing v_{Ca} term for Equation 1a.

8. The term (a_{TDM} Np63(t) e^{-\beta t}) does not seem to be the right fit for the process and, consequently, the memory effects described here. The terms seem to be reducing the influence of Np63 and Stat3 on TDM expression over time rather than the methylation of their promoters by Np63. This could also explain why the TDM curve seems to flatten out.

9. The authors mention that terminal differentiation is impaired upon increased pathogen load in Line 81. However, their model predicts a decreased threshold for differentiation on increased NFkB, a proxy for pathogen load, and this would imply ease of differentiation. The authors then contradict themselves by mentioning that keratinocyte differentiation is more sensitive in Line 268.

10. There were a few places where the references were not linked properly (Line 146, 157).

11. The GitHub link provided is accessible and has a README, but a better organization of the codes could be done along with steps on running the codes.

Reviewer #2: In this study, the authors develop a mechanistic model of keratinocyte differentiation under skin barrier damage and infection by constructing a regulatory network through the systematic curation of experimental study results. Furthermore, they identify a minimal regulatory network, termed the keratinocyte differentiation motif, using the kernel reduction method, and propose a dynamical system of equations to analyze the kinetics of this motif. The model’s predictions are subsequently validated using experimental data from in vitro keratinocyte differentiation assays. Their study reveals interesting results such as immune response from infections lowers the threshold Ca levels required for keratinocyte differentiation, etc. The modeling framework and the results presented in this manuscript are convincing. Therefore, I am in favor of the publication of this manuscript in PLOS Computational Biology. Below I am providing a series of comments throughout the manuscript, which I think will improve the manuscript.

Page 3, Line 77-79: This is a complex sentence conveying information about multiple processes and their causes. It is better to break it into simpler sentences to improve readability.

Page3, Line 81: Provide a brief explanation of skin barrier homeostasis here for readers who may not be familiar with this concept.

Page 3, Line 85: Would it be better to say "impact" rather than "contribute to", given that both factors negatively affect skin barrier homeostasis?

Page 3, Line 88-90: While this is partially true, there are experimental studies with in vivo animal models and in vitro epidermal or full-thickness skin models (Meesters et al., Keratinocyte signaling in atopic dermatitis: Investigations in organotypic skin models toward clinical application. Journal of Allergy and Clinical Immunology, 151(5), 1231 - 1235.)

Page 4, Line 118: The term "microenvironments" used here is not clearly defined. Does it refer to components such as the extracellular matrix, immune cells, signaling molecules, pathogens, or other factors?

Page 4, Figure 1(B): What is PKC? Is it Protein Kinase C? Need an explanation of this abbreviation.

Page 5, Line 141-143: It is nice to have a detailed description of various components of the regulatory network in the Supplementary material. However, it will be better to provide a brief description of the processes and components of the network here. It will help the readers to get an insight into this network and help the flow of the manuscript, without the need to go back and forth between the main text and SM.

Page 5, Line 146: Correct the cross-referencing here. There are a few others in the main text and in the SM, please check.

Page 5, Line 146-147: Why do you construct a Boolean model, rather than an agent-based model or ODE model? With biologically realistic parametrization, these models provide a realistic quantitative estimation for the dynamical variables. Including a brief explanation of the choice of Boolean model would be helpful.

Page 6, Line 171-173: How was the decision made regarding which node to remove in the kernel reduction algorithm? Was this choice based on the nature of the available data?

Page 6, Line 184: Is the steady-state line derived from experimental measurements, or does it represent the steady state from the model simulation?

Page 6, Figure 2(B): The y-axis of Figure 2(B) is a bit confusing. What does mean of normalized dynamics mean here?

Page 6, Figure 2(C): It seems to me that in Figure 2(C), the experimental data shows some sort of decay, whereas the model predicts some plateauing. It would be good to provide some statistical tests to make a quantitative assessment of how well the model’s predictions match the actual data.

Page 6, Figure 2(C): Similar to the previous comment, would adding more nodes (not reducing the network to just two nodes, but having three or more) improve the model’s ability to capture the experimental result.

Page 7, Equation (1a): Shouldn't the first term depend on the concentration of calcium as well?

Page 7, Line 202: What does maximal velocity signify here? Is it just the rate?

Page 7, Line 210-211: Provide a brief explanation of how the parameter estimates are obtained using this method.

Reviewer #3: This manuscript presents a mechanistic model of keratinocyte differentiation in response to skin barrier damage and infection.

This research provides insights into the mechanisms of skin barrier homeostasis maintenance under fluctuating environmental conditions.

The manuscript appears to be technically sound, presenting a well-structured approach to modeling keratinocyte differentiation in response to skin barrier damage and infection.

However, the manuscript could be improved if the authors add more information regarding several issues:

1.- the step-wise approach from Boolean to ODE modeling used in this manuscript has several potential weak points:

1.1- oversimplification in Boolean model, and potential loss of quantitative information: the Boolean model reduces complex interactions to binary on/off states, potentially oversimplifying the system's behavior. Similarly, limited temporal resolution: Boolean models typically use synchronous updates, which may not accurately capture the different timescales of biological processes

1.2- challenges in model reduction (potential loss of important interactions)

2.- potential limitations of the ODE model and parameter estimation uncertainty: the ODE model requires fitting to experimental data, which can be challenging given the limited availability of quantitative, time-resolved measurements.

3.- generalizability issues: the models' applicability to different experimental conditions or in vivo situations may be limited, as they are primarily based on and validated against specific in vitro experiments

It would be helpful if the authors add more details about the considerations taken to avoid the above issues.

Additionally, there are some issues with the writing quality:

- in the main text, and in suppl material, there are a number of "Error! Reference source not found" instances that should be corrected.

- there are instances of subject-verb agreement issues and awkward phrasing, such as "the outer skin called skin barrier".

- formatting inconsistencies: references to figures and sections are not consistently formatted throughout the text.

**Have the authors made all data and (if applicable) computational code underlying the findings in their manuscript fully available?**

Reviewer #1: Yes

Reviewer #2: Yes

Reviewer #3: Yes

PLOS authors have the option to publish the peer review history of their article (what does this mean? ). If published, this will include your full peer review and any attached files.

**Do you want your identity to be public for this peer review?** For information about this choice, including consent withdrawal, please see our Privacy Policy .

Reviewer #1: No

Reviewer #2: No

Reviewer #3: No

**Figure resubmission:**
---

## [Decision Letter · Decision Letter 1]

Dear Domínguez Hüttinger,

We are pleased to inform you that your manuscript 'History-dependent switch-like differentiation of keratinocytes in response to skin barrier damage' has been provisionally accepted for publication in PLOS Computational Biology.

Best regards,

Ricardo Martinez-Garcia

Academic Editor

PLOS Computational Biology

Stacey Finley

Section Editor

PLOS Computational Biology

Reviewer's Responses to Questions

**Comments to the Authors:**

Reviewer #1: The authors have addressed my comments.

Reviewer #2: I appreciate the authors' efforts in addressing my comments. I am satisfied with the revisions and support the acceptance of the manuscript.

Reviewer #3: I find the revised version satisfactory.

**Have the authors made all data and (if applicable) computational code underlying the findings in their manuscript fully available?**

Reviewer #1: None

Reviewer #2: Yes

Reviewer #3: Yes

PLOS authors have the option to publish the peer review history of their article (what does this mean? ). If published, this will include your full peer review and any attached files.

**Do you want your identity to be public for this peer review?** For information about this choice, including consent withdrawal, please see our Privacy Policy .

Reviewer #1: No

Reviewer #2: No

Reviewer #3: No

---

## [Editor Report · Acceptance letter]

PCOMPBIOL-D-24-01530R1

History-dependent switch-like differentiation of keratinocytes in response to skin barrier damage

Dear Dr Domínguez-Hüttinger,

I am pleased to inform you that your manuscript has been formally accepted for publication in PLOS Computational Biology. Your manuscript is now with our production department and you will be notified of the publication date in due course.

With kind regards,

Anita Estes
